# *In silico* guided structural and functional analysis of genes with potential involvement in resistance to coffee leaf rust: A functional marker based approach

**Geleta Dugassa Barka**[1,2], **Eveline Teixeira Caixeta**[1,3]*, **Sávio Siqueira Ferreira**[1¤],
**Laércio Zambolim**[1]

1 Laboratório de Biotecnologia do Cafeeiro (BIOCAFÉ), BIOAGRO, Universidade Federal de Viçosa (UFV), Viçosa, MG, Brazil, 2 Applied Biology Department, Adama Science and Technology University (ASTU), Adama, Oromia, Ethiopia, 3 Embrapa Café, Empresa Brasileira de Pesquisa Agropecuária, Brasília, DF, Brazil

¤ Current address: BioDiscovery Institute and Department of Biological Sciences, University of North Texas, Denton, TX, United States of America

* eveline.caixeta@embrapa.br

**Data Availability Statement:** The sequence data are available in the NCBI database (accession number KY485942.1).

## Abstract

Physiology-based differentiation of $S_H$ genes and *Hemileia vastatrix* races is the principal method employed for the characterization of coffee leaf rust resistance. Based on the gene-for-gene theory, nine major rust resistance genes ($S_H$1-9) have been proposed. However, these genes have not been characterized at the molecular level. Consequently, the lack of molecular data regarding rust resistance genes or candidates is a major bottleneck in coffee breeding. To address this issue, we screened a BAC library with resistance gene analogs (RGAs), identified RGAs, characterized and explored for any $S_H$ related candidate genes. Herein, we report the identification and characterization of a gene (gene 11), which shares conserved sequences with other $S_H$ genes and displays a characteristic polymorphic allele conferring different resistance phenotypes. Furthermore, comparative analysis of the two RGAs belonging to CC-NBS-LRR revealed more intense diversifying selection in tomato and grape genomes than in coffee. For the first time, the present study has unveiled novel insights into the molecular nature of the $S_H$ genes, thereby opening new avenues for coffee rust resistance molecular breeding. The characterized candidate RGA is of particular importance for further biological function analysis in coffee.

## Introduction

Coffee is one of the most valuable cash crops in many developing economies as it provides employment opportunities in cultivation, processing and marketing activities, thereby sustaining the livelihoods of millions around the world [1]. *H. vastatrix*, the causative agent of coffee leaf rust, accounts for one of the major threats to coffee production in almost every coffee producing region. Despite the release of some resistant coffee cultivars in recent years, coffee rust

**Funding:** This work was financially supported by TWAS-CNPq (Third World Academy of Science-Conselho Nacional de Desenvolvimento Científico e Tecnológico) to GDB, Brazilian Coffee Research and Development Consortium (Consórcio Brasileiro de Pesquisa e Desenvolvimento do Café – CBP&D/Café) to ETC and LZ, Foundation for Research Support of the state of Minas Gerais (FAPEMIG) to ETC and LZ, National Council of Scientific and Technological Development (CNPq) to ETC AND LZ, National Institutes of Science and Technology of Coffee (INCT/Café) to ETC and LZ, and Coordination for the Improvement of Higher Education Personnel (CAPES) to ETC and LZ. The funders had no role in study design, data collection and analysis, decision to publish, or preparation of the manuscript. URL: https://twas.org/opportunity/twas-cnpq-postgraduate-fellowship-programme URL: http://www.consorciopesquisacafe.com.br/ URL: http://fapemig.br/pt/ URL: http://cnpq.br/ URL: http://www.inctcafe.ufla.br/pt/ URL: https://www.capes.gov.br/.

**Competing interests:** The authors have declared that no competing interests exist.

continues to adversely affect coffee production and undermines the incomes of many households [2]. To date, at least 49 characterized physiological races of *H. vastatrix* have been reported [2,3]. The persistent emergence of new races and the sporadic outbreaks of this disease have imposed challenges in resistance breeding. The most pressing concern is, however, the breakdown of resistance genes leading to disease susceptibility of cultivars that were once validated as superior genetic material for resistance breeding [4].

The molecular profiles of coffee genes involved in different metabolic pathways, their evolution and annotation have been unveiled with the complete sequencing of *C. canephora* genome [5]. Given that *C. canephora* contributes to half of the Arabica coffee genome, being a natural hybrid of *C. canephora* and *C. eugenioides*, open access to its genome has provided valuable insights into the genome of *C. arabica* during the past five years. The discovery and successful introgression of $S_H3$ resistance gene locus to cultivated Arabica coffee from *C. liberica* was another landmark often considered as one of the greatest milestones in the development of coffee rust resistance [6]. Since then, molecular and physical mapping has enabled the sequencing and annotation of the $S_H3$ region, resulting in the discovery of multiple resistance (R) genes [6,7]. Dominantly inherited, the largest class of R-genes encode nucleotide-binding site leucine-rich-repeat (NBS-LRR) proteins that directly recognize the corresponding virulence (v) protein of the pathogen or its effects [8,9]. These genes are believed to contain several hundred gene families, which are unevenly distributed in the genomes of different plant species [10]. Intracellular signaling domain, similar to Drosophila toll/mammalian interleukin-1 receptor (TNL, Toll-NBS-LRR) and the coiled-coil (CNL, CC-NBS-LRR), are the two major N-terminal amino acid sequences preceding the NBS domain involved in specific signal transduction [8,11]. The other N-terminal domain linked to LRR includes leucine-zipper (a transmembrane protein, TM), protein kinase (PK) and WRKY TIR proteins [12]. These domains are predominantly involved in resistance signal transduction via conformational changes [13]. On the carboxyl-terminal region is the LRR, mediating specific protein-protein interaction to recognize pathogen effectors [14,15]. Although these domains are few in number, nucleotide polymorphism and variability of the LRR region are responsible for the perception of a specific pathogen effector [9,16]. Inter and intraspecific extreme variabilities of NBS-LRR have been attributed to gene duplication, unequal crossing over, recombination, deletion, point mutation and selection pressure due to continuous response to diverse pathogen races [6].

The readily available Arabica coffee BAC libraries constructed from disease resistant genotypes at different laboratories have accelerated studies involving resistance gene cloning [17,18]. Furthermore, the application of arbitrary DNA-based and functional (gene) markers in gene cloning has benefited crop improvement, either through map-based cloning using the former or direct gene cloning using the latter or both [19]. Direct cloning of the gene of interest over map-based gene cloning is appealing as this method is more precise and straightforward for gene characterization.

In coffee, reports on the origin and organization of disease resistance genes have begun to emerge in recent years as part of an effort to understand the role of major rust resistance genes. One such endeavor was the assembly of R genes spanning the $S_H3$ locus with the objective of tracing the evolution and diversity of LRR domains in three coffee species [6]. Despite the partial sequencing and annotation of several disease resistance genes in Arabica coffee [20], completely sequenced and characterized candidate genes are not yet readily available. Resistance to rust is conferred by nine major genes ($S_H1-9$) and the corresponding $v_{1-9}$ pathogen factors are known for long in the coffee rust pathosystem [3,21]. Nonetheless, molecular and functional characterization of any of the $S_H$ genes and the associated regulatory elements is entirely obscure, yet holds immense potential in changing the perspective of rust resistance breeding. Likewise, the use of functional markers that serve as a direct rust resistance screening

tool amongst the differential coffee clones is important but is barely addressed. The lack of a typical candidate rust resistance gene is one of the bottlenecks in coffee breeding. A resistance gene analog (RGA) marker, CARF005, was previously confirmed to share disease resistance ORF region in coffee [20,22]. This polymorphic RGA marker encodes the disease resistance protein domain NB-ARC (nucleotide binding site-ARC: ARC for APAF-1, R protein and CED-4) [23], exclusive in coffee cultivars resistant to *H. vastatrix* [22]. The complete sequencing and molecular characterization would help identify candidate disease resistance genes. The state-of-the-art bioinformatics analysis, availability of differential coffee clones with specific $S_H$ genes, structural and functional analysis of conserved domains and associated motifs of candidate RGAs belonging to the $S_H$ gene series could greatly advance coffee rust resistance breeding. Therefore, the objectives of this study were to trace the origin of resistance gene analogs (RGAs) involved in coffee rust resistance and perform comparative molecular characterization of selected candidate gene to determine whether it belongs to the $S_H$ gene series. We also investigated if any of the RGAs were activated during incompatible interaction between C. *Arabica* and *H. vastatrix*.

## Materials and methods

### Plant materials

Twenty-one differential coffee clones containing at least one of the coffee rust resistance genes ($S_H$1-9) and three genotypes susceptible to all the virulence factors (v1-9) of *H. vastatrix* were used in the CARF005 screening. The differential clones were initially characterized by CIFC (Centro de Investigação das Ferrugens do Cafeeiro, Portugal) for the identification of the different physiological races of *H. vastatrix*. All clones were vegetatively propagated at the Plant Pathology Department greenhouse of the Universidade Federal de Viçosa (Brazil). Genomic DNA was extracted from a young, second pair of leaves following the protocol developed by Diniz et al. [24]. DNA integrity was checked by electrophoresing in 1% gel and visualized after staining with ethidium bromide (0.5 μg/ml), followed by quality and quantity check of the extracted DNA by Nanodrop (NanoDrop Technologies, Wilmington, USA). DNA was stored at -20˚C until further use. RNA-Seq libraries from the work of Florez et al. [25] (hereafter, referred to as transcriptome), which comprise samples collected at 12 and 24 h after infection (hai) of *C. arabica* CIFC 832/2 with *H. vastatrix* (race XXXIII), were used as a reference in the search for novel candidate resistance genes.

### PCR conditions

A Sigma made (Sigma-Aldrich, Belo Horizonte, Brazil) disease RGA primer pair, CARF005, (F: 5′-GGACATCAACACCAACCTC-3′ and R: 5′-ATCCCTACCATCCACTTCAAC-3′) [26] was used to screen the differential host clones. PCR reagents were 1x buffer, 0.2 mM dNTPs, 0.2 μM primers, 1 mM $MgCl_2$, 0.8 units of Taq polymerase (Invitrogen, Carlsbad, USA) to which 5 ng gDNA was added to form a reaction volume of 20 μl. PCR cycling parameters were as follows: DNA denaturation at 95˚C for 5 min followed by 35 cycles of 94˚C for 30 s, 60˚C for 30 s and 72˚C for 1 min, followed by an extension step at 72˚C for 10 min. PCR products were screened for target inserts by electrophoresing in 1% UltraPure™ agarose (Invitrogen) and visualized after staining with ethidium bromide (0.5 μg/ml). All PCR and gel electrophoresis conditions were maintained consistently throughout the study unless stated otherwise.

## Screening of BAC clone

BAC library comprising 56,832 clones, constructed using renowned rust resistant Hibrido de Timor clone CIFC 832/2 [18] was used as the target source for RGA (CARF005). These clones were replicated in 384-well titer plates using a plate replicator sterilized with 10% $H_2O_2$ for 2 min, rinsed in sterile water for 10 seconds and soaked in 70% ethanol in laminar airflow cabinet. After the alcohol evaporated (3–5 min), the old cultures were copied to a new 384-well titer plate containing 70 μl fresh LB media (supplemented with 12.5 μgml$^{-1}$ chloramphenicol) in each well. Culture multiplication was achieved by incubating the plates at 37˚C for 18 h and shaking at 180 rpm. The identification of clones using the CARF005 insert was performed by grouping and subsequent group decomposition of the 384 clones until a single clone was identified as outlined in S1 Fig. BAC DNA was extracted using the centrifugation protocol of Wizard® SV Plus Minipreps DNA Purification System (Promega, Fitchburg, USA).

## Sequencing and contig assembly

The single BAC clone isolated using the CARF005 fragment was sequenced using the Illumina HiSeq2000/2500 100PE (paired-end reads) platform at Macrogen (Seoul, South Korea). Paired-end sequence processing and contig assembly were performed using SPAdes software [27]. Contigs that matched bacterial genome *(E. coli)* and sequences of the flanking vector (pCC1BAC$^{TM}$) were excluded prior to any downstream sequence processing. The assembled BAC contigs were used to map the transcriptome constructed from coffee genes that were activated in response to *H. vastatrix* infection using Tophat 2 [28] to locate the contig region with active gene expression.

## Gene prediction and annotation

Contigs with ≥ 200 bp size and sharing ≥ 90% identity with *C. canephora* were subjected to Augustus gene prediction [29]. Among the available genomes in the Augustus dataset, *Solanum lycopersicum* was used as a reference genome, as they share common gene repertoires and have similar genome sizes [30]. The predicted ORFs were annotated using different online annotation tools. First, Conserved Domain Database (CDD) of the NCBI was used to detect the conserved domains and retrieve their description, followed by the use of Predict Protein Server tool [31] molecular analysis and associated GO search. Protein 3D structure and nucleotide (ATP/ADP/GTP/GDP) binding sites were predicted using I-TASSER suite online tool [32]. The Research Collaboratory for Structural Bioinformatics Protein Data Bank (RCSB PDB) was used to generate the top ten I-TASSER server threading templates identified by LOMETS, based on the default Z-score for the highest significance in the threading alignment for the ultimate generation of the lowest energy 3D structure, all based on default parameters. Accordingly, the first structural 3D model was selected among the generated alternatives based on TM-score of 0.5 or higher for TM-align structural alignment. Functional prediction (ligand binding sites) was performed using COFACTOR COACH default parameters. As an annotation complement, the predicted ORFs were queried against the coding sequences (CDS) of *S. lycopersicum* (Sol Genomics Network: https://solgenomics.net/tools/blast/) and *V. vinifera* (Phytozome 11: https://phytozome.jgi.doe.gov) genomes.

## Sequence alignment and comparative analysis

Genes encoding resistance proteins were mapped to the *C. canephora* genome [5] to trace their probable origin and organization. BLASTn program was used to query the obtained sequences against the *C. canephora* genome (http://coffee-genome.org/blast). Synteny map

was constructed using *C. canephora* and *C. arabica* genomes to reveal the relative positions of candidate RGAs. The transcriptome reads (differentially expressed against *H. vastatrix*) were mapped to contig 9 using Tophat2 (-N 3—read-gap-length 3—read-edit-dist 6—no-coverage-search —b2-very-sensitive) [28] to locate the region of the contig containing the genes encoded in response to pathogenicity. The intergenic physical position, distance and orientation were analyzed for the RGAs.

## Point mutation analysis

The RGAs were analyzed for indels and substitutions using the EMBL MUSCLE multiple sequence alignment tool (http://www.ebi.ac.uk/Tools/msa/muscle/) and MEGA7[33] (www.megasoftware.net). Gene duplication was exclusively analyzed using MEGA 7, while DNA polymorphism and non-synonymous/synonymous substitution rates (ka/ks) were analyzed using DnaSP v5.1 [34].

## Functional and phylogenetic analysis

Based on the molecular evolution of protein domains, functional diversity between two NBS-LRR RGAs from coffee was analyzed. Homology was compared for the two RGAs in order to identify orthologous genes in the genomes of *S. lycopersicum* (https://solgenomics.net/tools/blast/) and *V. vinifera* (https://phytozome.jgi.doe.gov/pz/portal.html). Subsequently, comparative phylogenetic analysis of six amino acid sequences of the two NBS-LRR RGAs (predicted in the present study) and accessed from the two related genome databases was performed. Amino acid sequence positions with gaps and missing data were not considered for the inclusion of 554 dataset positions to generate phylogenetic tree using MEGA7 [33]. The evolutionary history was inferred using the minimum evolutionary method [35].

# Results

## Resistance gene screening among the differential coffee clones

To investigate the linkage of RGAs to known $S_H$ genes, differential coffee clones with different $S_H$ genes were subjected to RGA screening using the functional marker, CARF005. Of the 21 differential coffee clones and the three coffee genotypes susceptible to all known races of *H. vastatrix* (used as negative control for CARF005 marker), the marker was detected in eight clones as presented in Table 1 and S2 Fig. All clones with the $S_H$6 gene had the marker, while those without the gene failed to amplify by the PCR. Thus, gel analysis of the PCR amplicon revealed that this particular RGA marker amplified the $S_H$6 gene locus; however, two exceptions were observed. One of them was that the gene's allele was detected in CIFC 128/2-Dilla & Alghe, which is supposed to have just the $S_H$1 gene. In addition, CARF005 was found to be amplified in the differential clone CIFC 644/18 H. Kawisari, for which no $S_H$ gene has been reported to date.

## Sequence analysis of ORFs

Identification of a BAC clone using CARF005 and the comparative analysis of the RGAs with the other ORFs from *C. canephora* was performed to localize their relative position and determine putative function. To characterize genetic loci conferring resistance to leaf rust, a BAC clone 78-K-10 (with ~146 kb insert) was identified as shown in S1B & S1C Fig. Illumina HiSeq2000/2500 100PE generated 8,711,320 reads. After removing vector and noisy sequences, quality sequences (>20 QC) were assembled into 86 contigs of ≥ 200bp from which the two contigs, contigs 3 (16570 bp) and 9 (8285 bp), were selected (as they had >90% similarity with

**Table 1. $S_H$ gene allelic polymorphism detection in 22 differential coffee clones using CARF005 marker.**

| No. | Differential clone* | Susceptible to (*H. vastatrix* physiological race) | $S_H$ gene conferred** | Allelic difference (+/-) |
|---|---|---|---|---|
| 1 | 832/1- Híbrido Timor | - | 6,7,8,9,? | + |
| 2 | HW17/12 | XVI,XXIII | 1,2,4,5 | - |
| 3 | 1343/269- Híbrido Timor | XXII,XXV,XXVI,XXVII,XXVIII,XXIX, XXXI,XXXII,XXXIII,XXXVII,XXXIX,XL | 6 | + |
| 4 | H153/2 | XII, XVI | 1,3,5 | - |
| 5 | H420/10 | XXIX | 5,6,7,9 | + |
| 6 | 110/5-S 4 Agaro | X,XIV,XV, XVI,XXIII,XXIV,XXVI, XXVIII | 4,5 | - |
| 7 | 128/2-Dilla and Alghe | III, X, XII, XVI, XVII, XIX,XXIII, XXVII | 1 | + |
| 8 | 134/4-S12 Kaffa | X, XVI, XIX, XX, XXIII, XXVII, | 1,4 | - |
| 9 | H419/20 | XXIX, XXXI | 5,6,9 | + |
| 10 | 635/3-S 12 Kaffa | X, XIV,XV,XVI,XIX, XXIII,XXIV,XXVI,XXVII,XXVIII | 1,4,5 | - |
| 11 | 87/1-Geisha | III, X, XII, XVI, XVII, XXIII | 1,5 | - |
| 12 | 1006/10-KP 532 | XII,XVI,XVII, XXIII | 1,2,5 | - |
| 13 | 7963/117-Catimor | XXXIII | 5,7 or 5,7,9 | - |
| 14 | H420/2 | XXIX, XXX | 5,8 | - |
| 15 | 4106 | - | 5,6,7,8,9,? | + |
| 16 | 644/18 H. Kawisari | XIII | ? | + |
| 17 | 832/2- Híbrido Timor | - | 6,7,8,9,? | + |
| 18 | H147/1 | XIV, XVI | 2,3,4,5 | - |
| 19 | 32/1-DK1/6 | I,VIII, XII, XIV, XVI, XVII, XXIII,XXIV, XXV, XXVIII, XXXI | 2,5 | - |
| 20 | H152/3 | XIV,XVI, XXIII, XXIV, XXVII | 2,4,5 | - |
| 21 | 33/1-S.288-23 | VII, VIII, XII, XIV,XVI, | 3,5 | - |
| 22 | Caturra (c) | All | 5 | - |
| 23 | Catuaí 2143–236 (c) | All | 5 | - |
| 24 | Mundo Novo -376/4 (c) | All | 5 | - |

*Differential clones were from CIFC (Centro de Investigação das Ferrugens do Cafeeiro, Portugal).

**$S_H$1-9 genes as inferred by CIFC. Unknown race (-), coffee genotypes used as negative control (c), presence/absence of allelic differences among $S_H$ genes (+/-) and unknown $S_H$ gene (s) (?).

*Coffea canephora* DNA sequence) and then subjected to downstream processing. The sequences of the two contigs were combined and deposited at NCBI (accession number: KY485942.1). These contigs shared sequence identity with *C. canephora* contigs at different chromosomal regions, with the highest identity (99% for contig3 and 97% for contig 9) being on chromosome 0. All the 13 ORFs predicted (eight in contig 3 and five in contig 9) had matched to different species when queried against protein database of NCBI or to the *C. canephora* genome hub with significant similarities ($\leq$ 1e$^{-05}$ e-value) when BLASTn was used as presented in S1 Table. Among these, five genes (genes 5, 9, 10, 11 and 12) shared significant identities with RGAs from *C. canephora*. These genes are homologous to sequences in the *C. canephora* genome with the highest query coverage being on chromosome 0 as presented in Table 2. Genes 5 (intron-less, 1130 aa) and 11 (with two introns and two exons, 1118 aa) were the largest genes predicted. Both genes were located on the negative reading frames that belong to the CC-NBS-LRR gene family. BLASTn against the *C. canephora* genome showed that these genes are separated by 1,634,522 bp, although they are delimited with a shorter length (460 bp) in *C. arabica*. Synteny mapping of contig 9 to *C. canephora* and *C. arabica*, on the other hand, resulted in the localization of genes 9, 11 and 12, but with low coverage for gene 10 in the two

**Table 2. Size and structure of five resistance gene analogs and their mapping to chromosome 0 of *C. canephora* genome.**

|  | Genes* | | | | |
|---|---|---|---|---|---|
|  | 5 | 9 | 10 | 11 | 12 |
| Contig | 3 | 9 | 9 | 9 | 9 |
| Exon 1 | 3393 | 113 | 121 | 1175 | 345 |
| Intron 1 | - | 554 | 87 | 611 | 1786 |
| Exon 2 | - | 118 | 112 | 2222 | 183 |
| Intron 2 | - | 121 | 711 | 124 | - |
| Exon 3 | - | 69 | 121 | - | - |
| Intron 3 | - | 91 | - | - | - |
| Exon 4 | - | 155 | - | - | - |
| Intron 4 | - | 476 | - | - | - |
| Exon 5 | - | 130 | - | - | - |
| Query coverage (%) | 99.94 | 72.68 | 30.48 | 99.46 | 97.33 |
| Identity (%) | 76.00 | 85.00 | 79.00 | 68.84 | 73.00 |
| E-value | 0.00 | 9,00E-30 | 5,00E-17 | 0.00 | 3,00E-48 |
| Frame | N | N | P | N | P |
| Start hit-End hit | 108638370–108641761 | 106998076–106999730 | 107000654–107000761 | 107000357–107003848 | 107000234–107004551 |
| Protein (aa) | 1130 | 194 | 117 | 1118 | 175 |

*Exon and intron sizes are in nucleotides. N, negative reading frame and P, positive reading frame. Gene prediction was performed by Augustus command-line version gene prediction [29].

genomes (Fig 1). This analysis showed that genes in contig 9 are highly syntenic among HDT, *C. canephora* and *C. arabica* genomes.

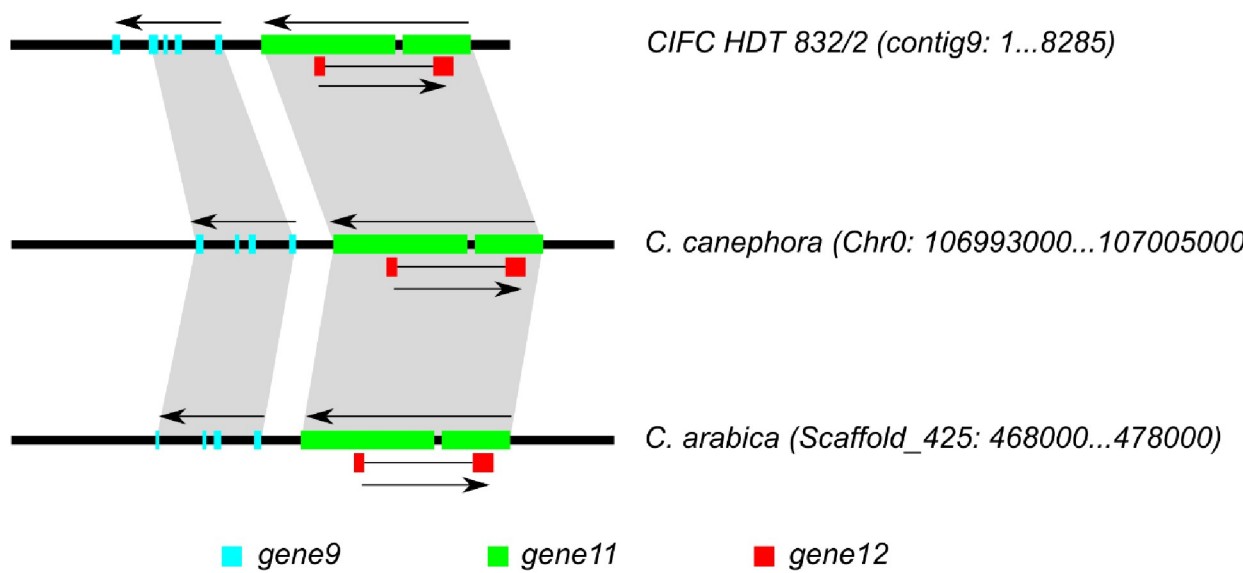

**Fig 1. Synteny analysis among CIFC HDT 832/2 contig 9, *C. canephora* and *C. arabica* genomes.** Contig 9 harbors gene 9, gene 10, gene 11, and gene 12 and their relative positions in the other genomes are shown by gray shading. Gene 10 was excluded from the figure due to its low coverage in the other genomes (Table 2). Black bars and colored boxes represent species genomes and genes (exons), respectively. Arrows indicate the gene orientation. Sequences from *C. canephora* (v1.0) and *C. arabica* (UCDv0.5) genomes were retrieved from Coffee Genome Hub (http://coffee-genome.org/coffeacanephora) and Phytozome 12 (https://phytozome.jgi.doe.gov/), respectively. Genomic positions of each gene were retrieved using BLASTn searches in the respective coffee genome portals, and then the figure was generated (in scale) using Inkscape software v0.94.2 (https://inkscape.org.

## CARF005 amplicon verification

Web-based PCR analysis was conducted to validate the specificity of the CARF005 primer pair and to complement the PCR amplification experiments. *In silico* PCR analysis using contig 9 and gene 11 ORF as the template strands, indicated that the CARF005 marker had a size of 400 bp (http://www.bioinformatics.org/sms2/pcr_products.html). This size of the amplicon was confirmed by PCR using the template DNA from the clone 78-K-10 as outlined in S1C Fig. Notably, the amplicon spans from base 6867 to base 7266 of contig 9 (8285 bp) and from base 369 to base 768 of gene 11 ORF (3354 bp) in a negative orientation, respectively.

## Gene annotation

Gene annotation and functional comparison were performed for genes 5 and 11 to reveal their molecular function, biological process and cellular localization associated GO terms (Table 3). The homology search for 13 ORFs, on the other hand, showed a range of protein arrays most of which had no role in disease resistance and lacked conserved domains (S1 Table). Among the five RGAs detected in either NCBI BLASTp or BLASTn against the *C. canephora* genome; genes 9 (unnamed protein product), 10 (putative resistance gene) and 12 (putative resistance gene) had similarity to RGAs as observed by mapping to *C. canephora* genome. Yet, genes 5 and 11 encode the largest resistance proteins (Gene 5: 126.81 kDa and pi: 7.65; gene 11:126.67 kDa and pi: 8.44), with many resistance-associated GO terms characterizing multiple functional domains.

## Gene characterization

To identify the candidate R genes activated against *H. vastatrix* incursion, two contigs (contigs 3 and 9) were mapped against the transcriptome of *C. arabica-H. vastatrix* interaction [25]. The number of reads that were mapped to RGAs spanning contig 9 showed that, after inoculation (12 and 24 hai), it was clear that a greater number of reads were mapped to a region of transcription hotspot (S3 Fig). Contig 3, from which gene 5 was predicted, was also mapped against the same transcriptome resulting in no matching transcripts that were differentially expressed at the two time points (12 and 24 hai) following pathogen inoculation. We predicted the likely region in contig 9 where most R genes are positioned. The result showed approximately 81.58% of the contig encodes transcripts of varying lengths associated with rust resistance, which are activated at 12 and 24 hai in response to *H. vastatrix* inoculation. Further analysis of genes 5 and 11 revealed that they belong to the NBS-LRR gene family (the major R genes in plants), suggesting the importance of continuing the *in silico* comparative structural and functional analysis. Intriguingly, both have the Rx_N superfamily (amino acid interval 10–96) characterized by N-terminal domain which is found in many plant resistance proteins and three (in gene 5) and four (in gene 11) additional multi-domains featuring the entire protein sequence (Fig 2). These genes can be referred as CC-NBS-LRR, as they comprise the N-terminal CC and LRR C-terminal domains flanking the NBS on either side.

In addition, annotation of both genes indicates that they encode defense proteins involved in various biological defense as demonstrated in Table 3. Moreover, alignment of the two resistance proteins encoded by genes 5 and 11 showed conserved and variable protein binding regions (Fig 3). Notably, although these genes share 90.24% nucleotide identity, their amino acid sequence identity is only 80.03%. The possibility of substitution mutation events was considered in explaining the diversity. Accordingly, the overall amino acid diversity was attributed to non-synonymous substitution events (non-synonymous/synonymous ratio, ka/ks = 1.5913) in both genes. Further analysis of LRR region showed a higher rate of non-synonymous substitution mutation (ka/ks, non-synonymous/synonymous substitution ratio = 1.9660).

**Table 3. Annotation and functional comparison of gene 5 and 11.**

| GO ID | GO term | Reliability (%) | Gene 5 | Gene 11 |
|---|---|---|---|---|
| **Molecular function ontology** | | | | |
| GO:1901363 | Heterocyclic compound binding | 49 | ✓ | ✓ |
| GO:0000166 | Nucleotide binding | 49 | ✓ | ✓ |
| GO:0005488 | Binding | 49 | ✓ | ✓ |
| GO:1901265 | Nucleoside phosphate binding | 49 | ✓ | ✓ |
| GO:0097159 | Organic cyclic compound binding | 49 | ✓ | ✓ |
| GO:0036094 | Small molecule binding | 49 | ✓ | ✓ |
| GO:0097367 | Carbohydrate derivative binding | 41 | ✓ | ✓ |
| GO:0017076 | Purine nucleotide binding | 41 | ✓ | ✓ |
| GO:0032559 | Adenyl ribonucleotide binding | 41 | ✓ | ✓ |
| GO:0032555 | Purine ribonucleotide binding | 41 | ✓ | ✓ |
| **Biological process ontology** | | | | |
| GO:0006952 | Defense response | 36 | ✓ | ✓ |
| GO:0006950 | Response to stress | 36 | ✓ | ✓ |
| GO:0050896 | Response to stimulus | 36 | ✓ | ✓ |
| GO:0002376 | Immune system process | 16 | ✓ | ✓ |
| GO:0006955 | Immune response | 16 | ✓ | ✓ |
| GO:0045087 | Innate immune response | 16 | ✓ | ✓ |
| GO:0044699 | Single-organism process | 14 | ✓ | ✓ |
| GO:0009987 | Cellular process | 14 | ✓ | ✓ |
| GO:0044763 | Single-organism cellular process | 14 | ✓ | ✓ |
| GO:0033554 | Cellular response to stress | 12 | ✓ | ✓ |
| GO:0016265 | Death | 12 | ✓ | ✓ |
| GO:0051716 | Cellular response to stimulus | 12 | ✓ | ✓ |
| GO:0012501 | Programmed cell death | 12 | ✓ | ✓ |
| GO:0008219 | Cell death | 12 | ✓ | ✓ |
| GO:0034050 | Host programmed cell death induced by symbiont | 12 | ✓ | ✓ |
| GO:0009626 | Plant-type hypersensitive response | 12 | ✓ | ✓ |
| GO:0009814 | Defense response, incompatible interaction | 7 | ✓ | ✓ |
| **Cellular component ontology** | | | | |
| GO:0016020 | Membrane | 33 | ✓ | ✓ |
| GO:0044464 | Cell part | 33 | ✓ | ✓ |
| GO:0005623 | Cell | 33 | ✓ | ✓ |
| GO:0005737 | Cytoplasm | 32 | ✓ | ✓ |
| GO:0044424 | Intracellular part | 32 | ✓ | ✓ |
| GO:0005886 | Plasma membrane | 31 | ✓ | ✓ |
| GO:0071944 | Cell periphery | 31 | ✓ | ✓ |
| GO:0043227 | Membrane-bounded organelle | 24 | ✓ | ✓ |
| GO:0043226 | Organelle | 24 | ✓ | ✓ |
| GO:0005634 | Nucleus | 24 | ✓ | ✓ |

Annotation was performed by Predict Protein online server [31] (URL: https://www.predictprotein.org).

## Comparative analysis of structural and functional sites

Structural modeling and comparative analyses of the identified genes was performed in order to infer the possible protein functions. Therein, we found that the number of LRR domains in genes 5 and 11 is variable (11 and 13 repeats, respectively) and arranged differently. We noted

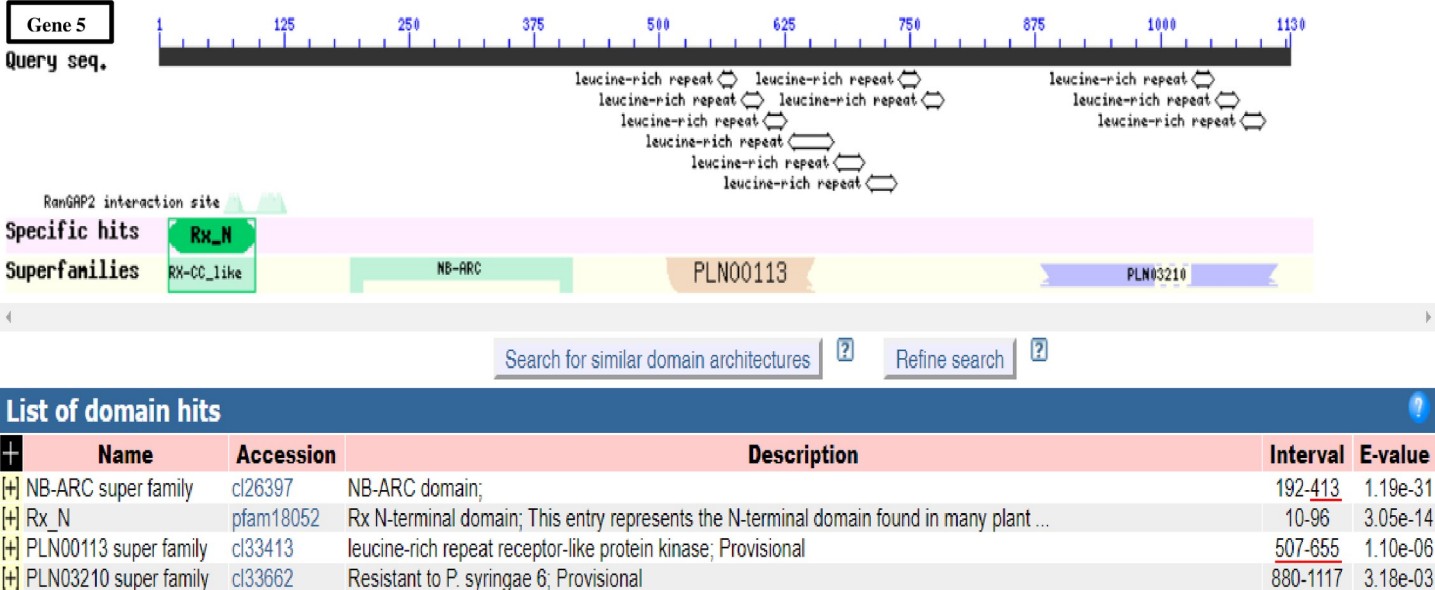

**Fig 2. Comparison of conserved domains and motif architecture in genes 5 and 11.** Different numbers of domain hits and variations in the domain length were detected and compared in both genes. Red bar (s) on the left side was (were) used to spot domain (s) exclusive to each gene while those on the right side were used to show the variation in the interval (number) of the amino acids constituting the respective domain and the polymorphisms in the size of the domains. Conserved domains were detected using NCBI Conserved Domain Database https://www.ncbi.nlm.nih.gov/Structure/cdd/wrpsb.cgi). Graphical summary was set to standard view.

the introduction of a coenzyme domain (CoaE, dephospho-CoA kinase) in gene 5, while LRR variants (LRR_8 and LRR_5) were introduced in gene 11 (Fig 2). Despite sharing similar protein multi-domains, two trans-membrane motifs (spanning 5–22 and 102–119 amino acid regions) were detected exclusively in the coiled-coil domain of gene 11 (Fig 4DII). The amino acid sequences of genes 5 and 11 were further analyzed for protein and nucleotide binding site polymorphism. Protein binding sites of the two genes revealed different sensitivity to substitution mutations. Few sites that were specific to each gene were highly affected while most of the binding sites had moderate to no effect as presented in S2 Table. The analysis revealed 15 protein binding sites in gene 5 and 7 sites in gene 11. Similarly, their secondary structures and solvent accessibility properties revealed more of conserved features (Fig 4AI-IV & 4CI-IV). Nevertheless, the amino acid residues forming the protein binding sites of the two genes showed high variability in the LRR regions (Figs 3, 4A & 4C). Although most of the residues are not conserved, the ADP binding site of the NBS domain contained some conserved sites (Fig 4BII & 4DII).

## Interlocus comparison of $S_H$ genes

To investigate the conserved regions in the five RGAs (genes 5, 9, 10, 11 and 12), contigs 3 and 9 were queried against three *C. canephora* and 10 *C. arabica* specific contigs assembled from BAC clones spanning $S_H$3 locus from the work of Ribas et al. [6]. All the 10 $S_H$3 contigs matched with contig 3 but with varying alignment lengths and identities as presented in S3 Table. Contig GU123898 and HQ696508 (both specific to *C. arabica*) had the highest number of matches to contig 9 (from which four clustered RGAs were predicted) with alignment lengths of 170 nts (77.647% identity and $1.57e^{-31}$ e-value) and 179 nts (76.536% identity and $1.21e^{-26}$ e-value), respectively. The closest contig (HQ696508) is located on the complementary

```
gene5_aa    MADAAVSATVKAVLGTVISIAADRVGMVLGVKAELERLGKTTATIQGFLADADEKMHSQG    60        gene5_aa    VSQEKGCGIEELGTLKYLKGSLEIRNLGLVKGKEAAKQAKLFEKPNLSRLVFKWESNL-S    718
gene11_aa   MADTVISATVEVVLGTVISIAADRIGMARGVKAELERLSKTAAMMQGFLADCDEKMHTRG    60        gene11_aa   VSQEKGCGIEELGTLKYLRGSLEIRNLGLVEGKDAAKQAKLFEKPNLSRLRLDFRRKRGH    676
            ***:.:***.:**********:**. ********.**:* :******.*****:.*                   ******************:***********:**:***************:.:.  :. .

gene5_aa    VRGWLKELEDEVFKADNVLDELHYHNLRQEVKYRNQPMKKKVCFFFSFFNAIGFSSSLAS   120        gene5_aa    QKSDNRDEDVLEGLQPHPKLEKLKIGSFMGNKFPQWLINLPKLVVLRIEDCGRCSELPAL   778
gene11_aa   VREWLKQLEDEVFKADNVLDELNYNNLRWDVKVRNQPMKKKVCFFFSFFSSIGFSSSLAS   120        gene11_aa   RKSDNCDEDVFEGLQPPNLQKLEIRYFMGTKFSQWLINLPKLVELWIEDCKRCSELPSL    736
            ** ***:.*************.*.:*** .*********** .:.:*********                    .****.****:***** :*:**:* **.**.******** * **:***** ****:*.*

gene5_aa    KIRDINTNLERINQQANELGLVRKHQKEADAAGATASRQTDSIVVPNVVGRAVDESKIVE   180        gene5_aa    GQLPSLKRLCLKRLENIRYVGDEFYGITTNE-----GSSRASGSSARRKFFPALEKLKV    833
gene11_aa   KIRDINTNLERINRQANELGLVRKHQKEANATGATTSRPTDSIVVPNVVGRAGDESKIVE   180        gene11_aa   GQLPSLKRLYLNKLENIRSIGDEFYGITTNEEGEEKGRSRASGSSTRRRKFFPALEELRV   796
            *************:***********:**.*:**:**:***********:********                  ********* *:.***** :*********** * ****.*****.:****...*

gene5_aa    MLLTPSERVVSVIPITGMGGLGKTTLAKSVYNNTKIVENFGIKSWVCVAREIKIVELFKL   240        gene5_aa    AFMENLAEWKDADQVRSTIGE--ADVFPMLRNFHIQSCPQLTALPCSCKILDVENCRNIT   891
gene11_aa   MLLTPSEKVVSVIPITGMGGLGKTTLAKSVYNNTKIDENFGIKSWVCVAREIKIVELFKL   240        gene11_aa   AYMKNLVEWKDADQVRSTIAEEAADVFPMLMDLSIQHCPQTTLPCSCKILDVQYCRNLT    856
            *******:*********************************:******************                *:*:**.***********.*  ***** . . :**.** *. ***********: ***:*

gene5_aa    ILESLPGTKVEVDGREAIVQEIRRKLGEKRFLLVLDDVWNRQWGLWNDFFTTLLGLSTIK   300        gene5_aa    SIKTSYGTACVERLGIYSCDNLRELPVDVFGLSLQCLTISCCPRLISLGVNGKKCPLRC-   950
gene11_aa   ILESLTRTKVEVDGRDAIVQEIRGKLGEKRFLLVLDDVWNCEQEFWSDFFTTLLGLSTTK   300        gene11_aa   SIKTGYGTASVEKLKIGCCNNLRELPEDVFGSSLQRLSIESCPRLISLGVNGKKCPLPCL   916
            *****.: **:****:****** *************** :. :* ***********                   ****.****.**:* * .*:***** **** ***.*:*.*************** *

gene5_aa    GSWCILTTRLEPVANAVPRHLQMND-PYFLGKLSDDACWSMLKEQVIAGEEVPQELEAIQ   359        gene5_aa    ----------------------RSLRSVWVVSCPNLVSFSLNLQETPSLEEFVLDDCPKL   988
gene11_aa   GSWCILTTRLQPVANAVPRHLQMNDGPYFLGKLSDDACWSILEKLVVAGEEVPNELEALK   360        gene11_aa   ERLSIQYCYGLTTISDKMFESCQSLRSLSVECCPNLVSFSLNLQETPSLEDFALLNCPKL   976
            **********:************** *************:*::: *:****:.***:.:                 :****  * .**************.* *:****

gene5_aa    EQILRRCDGLPLAASLIGGLLLNNRKEKWHCIVQESLLNEDQGEIDQILKVSFDHLSPPS   419        gene5_aa    IPHNFKGFAFATSLRKLAIGPFSSDDSSIDDFDWSGLRSASTLRELYLQGLPRSKSLPHQ   1048
gene11_aa   KQILKKCDGLPLAAKLIG------------------------------------------   378        gene11_aa   IPHRFNGFAFATSLRNLWIGPFSSDDSSIDGFDWSGLRSASTLCKVHLEGLCHSDSLPHQ   1036
            :***:.******** ***                                                         ***.*.*********:* *********** .*********** ::.*:** .*.* *****

gene5_aa    VKKCFAYCSIFPQDTKLGEDELIELWVAEGFVLPDRENTGMIEERGGEYLRILLQSSLLE   479        gene5_aa    LQYLATLTSLSLADFGGIEVLPDWIGNLVSLETLELSDCRKLQSLPSEAAMRRLTKLTHV   1108
gene11_aa   VKKCFAYCSIFPQDTELGEDELIEHWVAEGFVLPDQKNTRMMEETGGEYLRILLQNSLLE   438        gene11_aa   LQYLTTLTSLNLKNFGRIEVLPDWIGNLVSLETLQLSNCEKLRCLPSEAAMRRLTKLTSV   1096
            ***************:********.**********::** *:***.***********.****                ****:*****.*..** **************** **.*.**  .**************.*

gene5_aa    KVADEGRTYYKMHDLVHDFAKSVLNPKSSSQDRYLALHSYEEMAENVRRNKAASIRSLFL   539        gene5_aa    QVDGCPLLRQRYSPQRGIYLEE    1130
gene11_aa   KVQDKLRTYYKMHDLVHDFAKSILNPESSNQDRYLALNSSEGLVEKTTMTIPASIRTLFL   498        gene11_aa   EVRRCPLLRQRYTPQRGIYLEE    1118
            ** *: .***************:***:**.******:* *  :.*.: *:***:***                    :* .*********.*********

gene5_aa    HSGGGISADMNMLSRFKHLHVLKLSGYDVVFLPSSIGKLLRLRLLDISSSGITSLPESLC   599
gene11_aa   HLEDGISAG--MLLRFKYLHVLRLSGNDVVFLPSSIGKLLHLRLLDISSSRIKSLPESLC   556
            *  .****.  ** ***:****:***.************:*********.* .*******

gene5_aa    KLYNLQTLTIGGYALEGGFPKRMSDLISLRHLNYYHDDTEFKMLVQIGRLTCLQTLEFFN   659
gene11_aa   KLYNLQTLTIRNNALGEGFPKRMNDLISLRHLNYYHHRAKFKMPMQMGQLTCLQTLKFFN   616
            **********  .**.*******.***********:  :.*** :*:*:*******:***
```

**Fig 3. Alignment of proteins encoded by genes 5 and 11 and the protein binding regions.** *In silico* prediction of protein binding regions of gene 5 (underlined in red) and protein binding regions of gene 11 (underlined in green) were shown. Conserved and variable residues of the protein binding regions were highlighted in the S2 Table. Amino acid substitution: unrelated amino acid substitution (space), weakly similar substitution (period), strongly similar substitution (colon) and conserved amino acids (star). Note the six indel or non-synonymous substitution resulting in the polymorphism of protein binding sites (blue encircled) in either of the sequence. Sequence alignment was carried out using Clustal Omega (http://www.ebi.ac.uk/Tools/msa/clustalo/).

strand of gene 11 and is 505 bp upstream of the position where CARF005 forward primer annealed to gene 11.

## Phylogenetic analysis

In an attempt to discern the ancestral relationship of a set of sequences, phylogenetic analysis was performed. Accordingly, two resistance gene families (the NBS-LRR and non-NBS-LRR) were identified, completely sequenced and mapped to chromosome 0 of *C. canephora* genome with a query coverage of 99.94% for genes 5 and 11, 72.68% for gene 9, 33.05% for gene 10 and 97.52% for gene 12. The diversity of the NBS-LRR family was detected by analyzing the ka/ks ratios as presented in Table 4. The analysis revealed that the non-synonymous substitution event is common in the CDS, as revealed from all the pairwise analyses. Furthermore, the non-synonymous substitution of CDS is more prominent in the LRR region (in almost all pairwise comparisons).

Moreover, phylogenetic analysis showed that the tomato gene 5 was closely related to genes 5 and 11 of coffee than the gene 11 of both tomato and grape (Fig 5). Within coffee itself, a significant diversity between genes 5 and 11 was detected by the MEGA 7 bootstrap method of the phylogenetic analysis.

## Discussion

The majority of NBS-LRR encoding genes are known to be clustered but unevenly distributed in plant genomes [10,40–42]. The NBS-ARC domain is known to be involved in directly

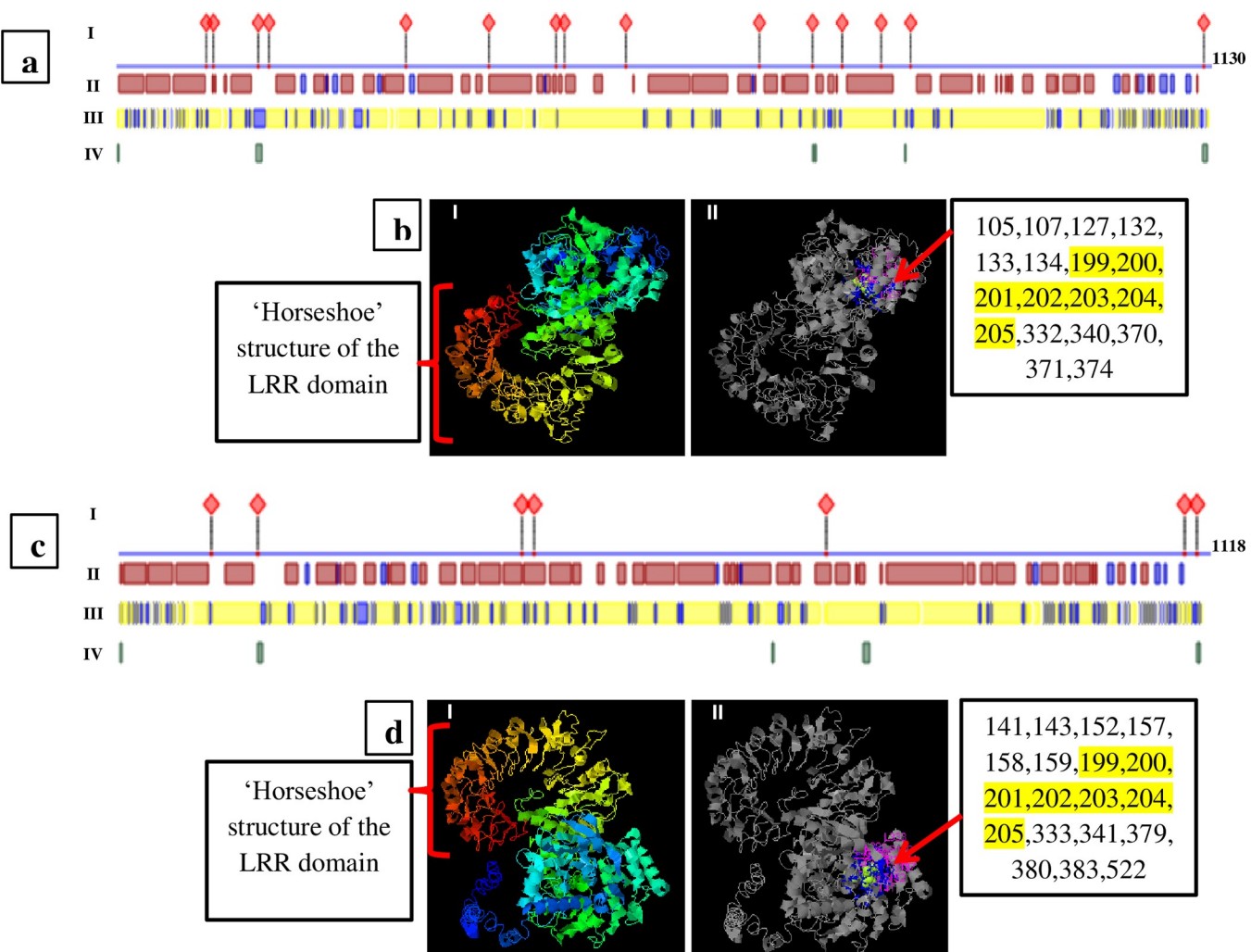

**Fig 4.** *In silico* **3D structure, protein and nucleotide binding site prediction for gene 5 (a and b) and 11 (c and d).** Protein binding and secondary structure (a and c): Protein binding sites (I), the three types of secondary structures are assumed at different regions (helical: red boxes, strand: blue boxes and loop: intervening white spaces) (II), solvent accessibility (exposed: blue boxes, buried: yellow boxes and intermediate: white spaces) (III) and high protein disorder and flexibility: green boxes (IV) [31] (https://open.predictprotein.org/). 3D structure and nucleotide binding sites (b and d): 3D structures with Rx-CC-like (blue) to LRR (red through yellow forming the 'horseshoe' structure) domains (bI & dI) and the colored residues (NBS) forming the nucleotide (ATP/GTP/ADP/GDP)-binding sites (bII and dII) of genes 5 and 11, respectively. Nucleotide binding site residues with the highest C-score are listed in the right box (conserved residues highlighted in yellow) with the red arrow indicating the sites. I-TASSER modelling [32] (https://zhanglab.ccmb.med.umich.edu/I-TASSER/) C-score was -1.73 and -1.75 (C-score ranging from -5 to 2, where 2 refers to the highest confidence) and 0.29 and 0.17 (C-score ranging from 0–1, where higher score indicates reliable prediction) for nucleotide binding prediction for the two proteins, respectively.

blocking the biotrophic pathogens by activating the hypersensitive response (HR) [43]. HR starts with programmed cell death of affected and surrounding cells and ends with the activation of systemic acquired resistance (SAR), in which the defense is induced in distal non-infected cells of the host under attack [44,45]. By recognizing the corresponding virulence (vr) factors or their effects, NBS-LRR proteins are sufficient to induce HR [8,9,45,46]. In the present study, a cluster of two different classes of RGAs resistant to coffee rust, the NBS-LRRs linked to non-NBS-LRR genes were reported. The two NBS-LRR genes (genes 5 and 11) are the largest non-TNL genes sequenced in Arabica coffee and most other plants investigated to date [6,47–49].

**Table 4. Pair-wise synonymous and non-synonymous nucleotide substitution analysis among the six resistance gene analogs (gene 5 and 11 and their respective two top hits as mined by BLASTn in NCBI).**

| Seq. 1 | Seq. 2 | Entire protein | | | LRR region | | |
|---|---|---|---|---|---|---|---|
| | | Ks | Ka | ka/ks | Ks | Ka | ka/ks |
| gene5_hit1 | gene11_hit1 | 0.0786 | 0.1302 | 1.6565 | 0.0702 | 0.1383 | 1.9701 |
| gene5_hit1 | gene11_hit2 | 0.0899 | 0.1614 | 1.7953 | 0.0536 | 0.1408 | 2.6269 |
| gene5_hit1 | gene11 | 0.0723 | 0.1177 | 1.6279 | 0.0622 | 0.1233 | 1.9823 |
| gene5_hit1 | gene5_hit2 | 0.0635 | 0.0999 | 1.5732 | 0.0583 | 0.1029 | 1.7650 |
| gene5_hit1 | gene5 | 0.0009 | 0.0039 | 4.3333 | 0.0015 | 0.0045 | 3.0000 |
| gene11_hit1 | gene11_hit2 | 0.1092 | 0.1854 | 1.6978 | 0.0756 | 0.1602 | 2.1190 |
| gene11_hit1 | gene11 | 0.0061 | 0.0164 | 2.6885 | 0.0095 | 0.0170 | 1.7895 |
| gene11_hit1 | gene5_hit2 | 0.0761 | 0.1309 | 1.7201 | 0.0736 | 0.1369 | 1.8601 |
| gene11_hit1 | gene5 | 0.0786 | 0.1288 | 1.6387 | 0.0701 | 0.1383 | 1.9729 |
| gene11_hit2 | gene11 | 0.1074 | 0.1742 | 1.6220 | 0.0686 | 0.1445 | 2.1064 |
| gene11_hit2 | gene5_hit2 | 0.0846 | 0.5829 | 6.8901 | 0.0607 | 0.1440 | 2.3723 |
| gene11_hit2 | gene5 | 0.0902 | 0.1620 | 1.7960 | 0.0540 | 0.1430 | 2.6481 |
| gene11 | gene5_hit2 | 0.0704 | 0.1155 | 1.6406 | 0.0616 | 0.1199 | 1.9464 |
| gene11 | gene5 | 0.0723 | 0.1164 | 1.6100 | 0.0622 | 0.1234 | 1.9839 |
| gene5_hit2 | gene5 | 0.0635 | 0.0997 | 1.5701 | 0.0583 | 0.1052 | 1.8045 |

Seq., sequence, non-synonymous/synonymous substitution rate was computed by DnaSP v5.1 [34].

We identified 13 genes: genes 1, 2, 3, 4, 5, 6, 7 and 8 (gene 5 is a RGA) from contig 3 and genes 9, 10, 11, 12, and 13 (only 13 is not a RGA) from contig 9 (Table 2; S1 Table). We also report, a completely sequenced and characterized novel RGA (gene 11), from Híbrido de Timor CIFC 832/2, probably a major component of the $S_H$ gene. This Híbrido de Timor (HDT) (*C. arabica* x *C. canephora*) is immune to all known virulence factors of *H. vastatrix* physiological races and therefore, is an extremely important source of resistance [50]. In addition to gene 11, mapping of the transcriptome from *C. arabica*-*H. vastatrix* interaction to contig 9 suggests that the three other clustered RGAs (genes 9, 10 and 12) were also differentially expressed during the incompatible interaction. It has also revealed the presence of reads exclusively mapped to transcripts of pathogen-infected plants at 12 and 24 hai (S3 Fig). Despite the identification of five RGAs from the two contigs, it was only the resistance gene locus spanning contig 9 which showed active expression of resistance transcripts against the *H. vastatrix* race. Therefore, this result suggests, qualitatively, that these genes might be differentially expressed during incompatible interactions at 12 and 24 hai with *H. vastatrix* (race XXXIII). What was more interesting was the difference in the resistance role of the two homolog RGAs (genes 5 and 11) belonging to the CC-NBS-LRR gene family, as seen by the differential expression analysis. In the present work, the unlikely role of gene 5 (no differential expression unlike gene 11) to confer resistance to *H. vastatrix* could be attributed to any of the indel or nonsynonymous substitutions in response to selection pressure which could have resulted in the changes to the protein binding sites as described elsewhere (Figs 3, 4AI & 4CI and S2 Table). A more illustrative molecular mechanism as to why gene 5 and 11 confer resistance to different pathogens could be explained by the 'decoy' and 'guarde' models, which propose the indirect recognition of pathogen effectors resulting in the neutralization of the complex by CC-NBS-LRRs proteins [51–54]. Owing to the enhanced diversifying selection pressure, the two genes could have resorted to recruiting different proteins (guardee and decoy)–to be further researched- that serve their respective protein binding domains for triggering differential expression.

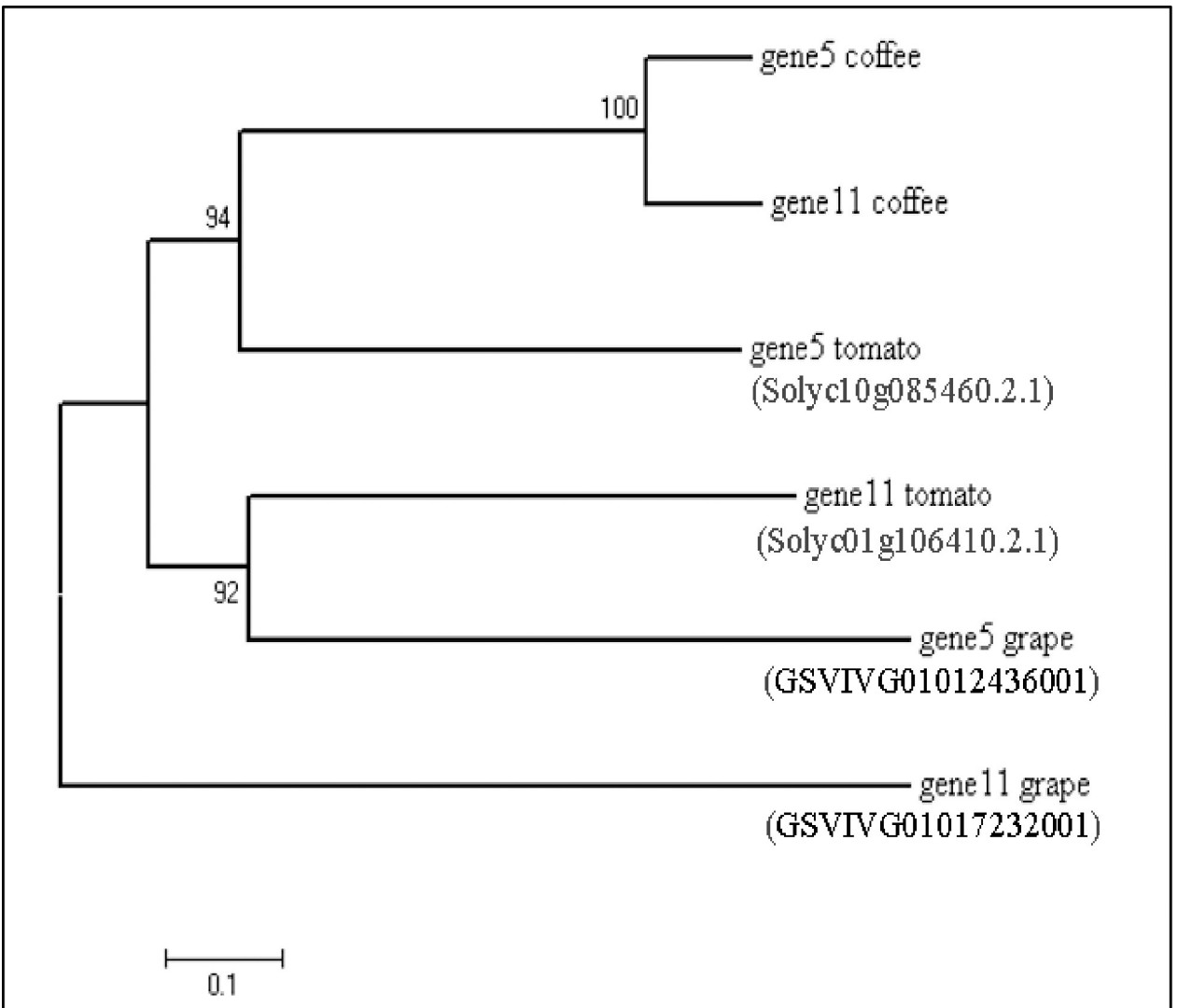

**Fig 5. Phylogenetic history of genes 5 and 11 in three related genomes.** The intensity of diversifying selection on the two CC-NBS-LRR encoding genes in the three related genomes showed the minimum such selection as an adaptive force for disease defense in coffee. The evolutionary history was inferred using the Minimum Evolution method [35]. The optimal tree with the sum of branch length = 2.98805978 is shown. The percentage of replicate trees with the associated taxa clustered together in the bootstrap test (500 replicates) are shown next to the branches [36]. The tree is drawn to scale, with branch lengths in the same units as those of the evolutionary distances used to infer the phylogenetic tree. The evolutionary distances were computed using the Poisson correction method [37] and are in the units of the number of amino acid substitutions per site. The ME tree was searched using the Close-Neighbor-Interchange (CNI) algorithm [38] at a search level of 1. The Neighbor-joining algorithm [39] was used to generate the initial tree. The analysis involved 6 amino acid sequences. All positions containing gaps and missing data were eliminated. A total of 554 positions were there in the final dataset. Evolutionary analyses were conducted using MEGA7 [33] (https://www.megasoftware.net/). Subject IDs are indicated in parenthesis for the corresponding two homologous sequences mined by BLASTx against tomato (Sol Genomics Network, https://solgenomics.net/) and grape (Phytozome, https://phytozome.jgi.doe.gov/pz/portal.html) genome databases.

The known rust resistance genes, $S_H3$ in *C. liberica* [55], $S_H6$, 7, 8 and 9 in *C. canephora* [56] and $S_H1$, 2, 4 and 5 genes in *C. arabica* are dominantly-inherited genes [57]. One of the fundamental questions to be clarified is how different are these 9 $S_H$ genes that belong to different coffee species. The comparative analysis of contigs from the $S_H3$ locus of *C. arabica* and *C. canephora* [6] revealed different levels of conservation of motifs in the two contigs examined: contigs 3 and 9. The results indicate that the RGAs may share large conserved regions, but few highly polymorphic regions encoding specific protein motifs necessary for critical

roles. This characteristic conservation of domains was once more confirmed based on comparative analysis of the cloned gene (gene 11) and using differential coffee clones for $S_H$ gene identification. PCR amplification of gene 11 also indicated the existence of allelic difference/polymorphism among the $S_H$ gene loci and considerable sequences of conserved domain (on which CARF005 primer was designed) with $S_H6$ and possibly with $S_H1$. PCR amplification using CARF005 primer (constituting gene 11) was detected in all the differential clones with $S_H6$ and 832/1-HT and 832/2-HT containing $S_H6$, 7, 8, 9 and $S_H$?. In addition, we report a conserved sequence of gene 11 in CIFC 128/2-Dilla & Alghe, previously considered to contain only the $S_H1$ gene, and in CIFC 644/18 H. Kawisari with an uncharacterized $S_H$-gene (Table 1). Overall, we propose the following hypothesis for extensive and rigorous biological investigation: the identified gene (gene 11) could be one of the unidentified and a not yet supplanted (at least in Brazil) $S_H$ gene in HDT consisting of a conserved domain (CARF005) shared with the $S_H6$ and $S_H1$ genes.

BLASTn of the RGAs against *C. canephora*, the result from differential clone screening and annotation altogether confirmed that gene 11 locus is descended from *C. canephora*, hence is a sibling of $S_H6$-9 [56]. Synteny mapping of the four RGAs (genes 9, 10, 11 and 12) to *C. canephora* and the recently sequenced Geisha variety (https://phytozome.jgi.doe.gov/pz/portal.html) showed conserved regions (Fig 1). These data were complemented by the differential clone screening for $S_H$ genes. Interestingly, conserved sequences (CARF005) were detected in eight of the differential clones, corroborating the existence of a strong linkage of the $S_H$ gene locus and the RGAs. The disparity of the position of gene 5 in relation to gene 9 (the fact that all the predicted genes are from an insert size of ~146 kbp) could be attributed to the nearby transposase gene (gene 1) (S1 Table). Transposons could have interrupted and separated the two genes apart by 1.6 Mb, since *C. arabica* is known to have diverged from *C. canephora*. Multiple transposable elements linked to NBS-LRR regions were reported in other plants [48,58]. Transposition of genes and gene fragments are some of the mechanisms that generate variability and positional changes among the NBS-LRR genes in different plants [48,58–61].

Rx-CC, PLN and NB-ARC domains are conserved in the NBS-LRR genes across diverse plant species [47,62,63]. The potato virus x resistance (Rx) protein-like N-terminal coiled-coil domain mediates intramolecular interaction with NB-ARC and intermolecular interactions through RanGAP2 (Ran-GTPase-activating protein-2) in potato [46,64]. Rx-CC, RanGAP2 interaction site and NB-ARC were detected in genes 5 and 11 (Fig 2), suggesting similarity in their defense role in coffee. However, unlike the Rx-CC domain with four helical structures, six helical structures are conserved in genes 5 and 11, indicating polymorphic differences between the species. The PLN00113 domain in gene 5 and PLN03210 in gene 11, span the LRR region and were initially reported in *A. thaliana* [47]. The distinct position of these domains in genes 5 and 11 indicates high variability in the LRR region in both genes. Functional motif prediction indicated that the PLN03210 (LRR domain) is likely engaged in direct effector interaction while the corresponding PLN00113 of gene 5 is engaged in LRR-reception and downstream kinase-mediated signaling. These observations are in accordance with the functional and structural analysis data of LRR proteins in *A. thaliana* [47,65–68]. Based of their annotations, the two proteins (encoded by genes 5 and 11) are intracellular resistance proteins that directly or indirectly recognize pathogen effector proteins and subsequently trigger a response that may be as severe as localized cell death [45].

Different selection pressures shape the evolution of domains in the NBS-LRR encoding genes. The NBS domain was assumed to be under the purifying selection (a negative selection in which variation is minimized by stabilizing selection) than by the diversifying selection, which acts on the LRR domain [9,69]. In contrast, the diversifying selection (positive selection) act on all the domains of genes 5 and 11 (ka/ks >1). This result is contrary to the general

assumption that diversifying selection is diluted when the overall non-synonymous substitution is considered [6], indicating an intense diversifying selection action on both genes. Further investigation of four more orthologous genes also resulted in similar findings, indicating that the NBS-LRR genes are highly variable due to substitution mutations. As the LRR domains are involved in direct ligand binding, their variability due to non-synonymous substitution is higher than that seen in other domains. This results in the formation of a super-polymorphic region to cope with the continuously evolving pathogen effectors. Similar findings (from different plants, including coffee) on diversifying selection have been reported [6,9,11,41,48,70,71]. Diversifying selection by non-synonymous substitution was also detected in non-NBS-LRR genes (genes 10 and 12, ka/ks = 4.86) and other crop plants [72,73], reiterating the importance of substitution mutation in such clustered R genes (Table 4). Synergistic activation of the two groups (NBS-LRR and non-NBS-LRR) may enhance the resistance durability; and so their expression pattern merits further investigation.

Based on the phylogenetic tree of orthologous genes originated from related genomes, the six genes could be divided into two groups (Fig 5). Gene 5 from tomato is closely related to genes 5 and 11 from coffee, making the first group, whereas genes 5 and 11 from grape happens to be the second highly diversified group. Intraspecies diversity of non-TIR-NBS-LRR due to substitution and genetic recombination exist in grape [74] and tomato [75] while gene duplication and conversion events were observed in coffee [6]. In general, the phylogenetic tree revealed that genes 5 and 11 may have recently diverged in coffee, while the divergence observed in the other species may have been older events.

## Conclusion

The two groups of RGAs, NBS-LRR and non-NBS-LRR, are clustered in a single locus from which multiple variants of resistance genes are expressed to confer specific resistance functions. The four cloned, sequenced and characterized RGAs span a rust resistance gene locus descended from *C. canephora*. The two CC-NBS-LRR protein encoding genes are under strong diversifying selection impacting all component domains. A more intense diversification of LRR region indicates that the variability in the effector binding site is the cause of divergence in resistance specificity. Although conserved sequences were detected for the $S_H6$ gene across the various differential coffee clones, it could be inferred that the $S_H$ gene loci have a characteristic polymorphism conferring different resistance phenotypes against coffee leaf rust. This is the first report unveiling new insights into the molecular nature of $S_H$ genes. The CC-NBS-LRR gene characterized is the largest and most complete sequence ever reported in Arabica coffee. The work demonstrated a cluster of resistance genes spanning the R gene locus that could serve as functional markers for subsequent functional analysis. These findings could also serve as a benchmark for validation of expression patterns in response to pathogenicity and gene segregation along generations. Such studies can be applied in molecular breeding as it has the potential to replace arbitrary DNA-based marker-assisted breeding at least for two reasons. First, there is no loss due to segregation, which is the case even for finely saturated markers. Second, four of the RGAs (genes 9, 10, 11 and 12) are stacked in a locus, from which different primers can be designed to screen genotypes to verify co-segregation analysis.

## Supporting information

**S1 Fig. Work flow in BAC clone screening.** Clone pooling and subsequent group decomposition to isolate a single clone with CARF005 insert (A), DNA of isolated clone 78-K-10 (B) and CARF005 PCR amplicon (C) as revealed by 1% UltraPure™ agarose gel electrophoresis. M is

100 bp DNA size marker. The red arrow indicates the estimated size of marker DNA.
(TIF)

**S2 Fig. The 21 differential coffee clones screened for CARF005 marker (listed in order as in Table 1).** Clones with CARF005 were 1 (832/1-HT), 3 (1343/269-HT), 5 (H420/10), 7 (128/2-Dilla and Alghe), 9 (H419/20), 15 (4106), 16 (644/18 H. Kawisan, a new report) and 17 (832/2-HT). M: DNA weight marker ladder (the lightest band being 100 bp). The last three lanes (22–24) represent three coffee genotypes susceptible to all known races of *H. vastatrix*, used in this experiment as negative control for the CARF005 marker gene. The red arrow indicates the estimated size of marker DNA. No gel cropping was performed to any of the lanes displayed.
(TIF)

**S3 Fig. Mapping of contig 9 to transcriptome of differentially expressed genes during *C. arabica-H. vastatrix* (race XXXIII) incompatible interaction to show the region of active gene (gene 9, 10, 11 and 12) expression.** Note the three expression profiles (three rows) corresponding to control (uninoculated at 0 hour, top row), 12 (middle row) and 24 hai (bottom row) of transcriptome reads mapped against contig 9 of resistant coffee clone (CIFC HDT 832/2). Grey shades indicate matching transcriptome reads while nucleotide substitutions (mismatches) were shown by colored strips (yellow: G, green: A, red: R and blue: C). Large red shades indicate deletions. The three RGAs presented in different colors were selected due to their higher coverage. The contrasting difference in the differential expression was quite clear between the control sample (0hai) and the two samples taken at the other time points (12 & 24hai) and remarkable difference in the number of activated transcripts of the genes at 12 and 24hai. Contig mapping was performed by Tophat 2[28] (http://ccb.jhu.edu/software/tophat) setting alignment parameter as '-N 3—read-gap-length 3—read-edit-dist 6—no-coverage-search —b2-very-sensitive' to locate the region of the contig encoding genes against the pathogen and visualized with Integrative Genomics Viewer (IGV) v. 2.3 [76] (http://www.broadinstitute.org/igv).
(TIFF)

**S1 Table. Top hits for the 13 ORFs as found in NCBI by BLASTp or at *C. canephora* genome by BLASTn.** *Homologous sequences for which no ID/Accession number has been assigned are indicated in hyphen. BLASTp was performed by NCBI online server (https://blast.ncbi.nlm.nih.gov/Blast.cgi?PAGE=Proteins).
(DOCX)

**S2 Table. Mutation (substitution) effect on protein binding regions of gene 5 and 11 indicated by amino acid sequence in respective genes.** *As numbered from the first to the last residue, the hyphen indicates the amino acid (s) constituting the binding sites. Purple highlighted residues are conserved residues in both genes while yellow highlighted residues are specific protein binding sites in respective gene. Substitution mutation effect analysis was performed by The Predict Protein Server [31] (http://ppopen.rostlab.org).
(DOCX)

**S3 Table. Output of the two contigs BLASTed against $S_H3$ locus contigs specific to *C. arabica* and *C. canephora** . *Ten contigs specific to *C. arabica* and three contigs specific to *C. canephora*, all assembled from BAC clones with $S_H3$ locus were taken from the work of Ribas et al. [6].
(DOCX)

## Acknowledgments

We thank Drs Jorge L.B. Pacheco, Abraham Abera, Bayissa Chala and Mohammed Naimuddin for their valuable suggestions and edition of the manuscript. We are also grateful to the Agronomic Institute of Paraná, Londrina-Brazil, for providing the CIFC 832/2 BAC library.

## Author Contributions

**Conceptualization:** Eveline Teixeira Caixeta.

**Data curation:** Sávio Siqueira Ferreira.

**Formal analysis:** Geleta Dugassa Barka, Sávio Siqueira Ferreira.

**Funding acquisition:** Geleta Dugassa Barka, Eveline Teixeira Caixeta, Laércio Zambolim.

**Investigation:** Geleta Dugassa Barka.

**Methodology:** Geleta Dugassa Barka, Sávio Siqueira Ferreira.

**Project administration:** Eveline Teixeira Caixeta, Laércio Zambolim.

**Resources:** Eveline Teixeira Caixeta, Laércio Zambolim.

**Software:** Geleta Dugassa Barka, Sávio Siqueira Ferreira.

**Supervision:** Eveline Teixeira Caixeta, Laércio Zambolim.

**Validation:** Geleta Dugassa Barka, Eveline Teixeira Caixeta, Sávio Siqueira Ferreira, Laércio Zambolim.

**Visualization:** Geleta Dugassa Barka, Sávio Siqueira Ferreira.

**Writing – original draft:** Geleta Dugassa Barka.

**Writing – review & editing:** Geleta Dugassa Barka, Eveline Teixeira Caixeta, Sávio Siqueira Ferreira, Laércio Zambolim.

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
