## [Decision Letter · Decision Letter 0]

11 Mar 2020

PONE-D-19-25123

Structural and functional analysis of genes with potential involvement in resistance to coffee leaf rust: a functional marker based approach

PLOS ONE

Dear Dr. Caixeta,

Thank you for submitting your manuscript to PLOS ONE. After careful consideration, we feel that it has merit but does not fully meet PLOS ONE’s publication criteria as it currently stands. Therefore, we invite you to submit a revised version of the manuscript that addresses the points raised during the review process.

We would appreciate receiving your revised manuscript by Apr 25 2020 11:59PM. To enhance the reproducibility of your results, we recommend that if applicable you deposit your laboratory protocols in protocols.io, where a protocol can be assigned its own identifier (DOI) such that it can be cited independently in the future. For instructions see: http://journals.plos.org/plosone/s/submission-guidelines#loc-laboratory-protocols

We look forward to receiving your revised manuscript.

Kind regards,

Binod Bihari Sahu, Ph.D.

Academic Editor

PLOS ONE

Journal Requirements:

Additional Editor Comments (if provided):

Please revise the ms addressing the concerns raised by the reviewer.

Reviewers' comments:

Reviewer's Responses to Questions

**Comments to the Author**

1. Is the manuscript technically sound, and do the data support the conclusions?

Reviewer #1: Yes

Reviewer #2: Yes

2. Has the statistical analysis been performed appropriately and rigorously? 

Reviewer #1: Yes

Reviewer #2: N/A

3. Have the authors made all data underlying the findings in their manuscript fully available?

Reviewer #1: Yes

Reviewer #2: No

4. Is the manuscript presented in an intelligible fashion and written in standard English?

Reviewer #1: Yes

Reviewer #2: Yes

5. Review Comments to the Author

Reviewer #1: The authors have described an important study relating to the disease resistance genes against coffee rust. The identification of two groups of RGAs, the NBS-LRR & non-NBS-LRR contained on the SH loci demonstrate polymorphism which is a new insight into molecular characterization of these genes.

Reviewer #2: The coffee is a cash crop and widely grown in different parts of the world. Coffee leaf rust (CLR) caused by an obligate fungal parasite Hemileia vastatrix is one of the major diseases costing millions of dollar loss to coffee growers. Identification of resistance gene and its utilization in the breeding program will be a substantial contribution to the coffee research. Here Authors have made an effort to identify resistance gene(s) for H. vastatrix by screening the BAC clone library of the resistance coffee crop, Hibrido de Timor (HDT), using the CARF005 marker. It is an innovative approach to identify resistance genes in coffee and hence worth pursuing. However, in the process of achieving the goal, the authors have fallen short at several points.

Major concerns:

• Authors have mentioned supplementary figure S5 in line no 273, Table S6 in line no 315, and table no S7 in line no 338, but these figures and tables were not provided to the reviewer to review.

• Also, the Authors mentioned about generating RNA-Seq libraries and its sequencing in line no 112 in material method sections. But detailed method of the RNA-Seq libraries preparation and sequencing are not provided anywhere. Authors need to confirm that if they have generated new RNA-Seq libraries for this study or cross-referenced to Florez et al., 2017. if authors have performed RNA-Seq for this study, then they should submit the sequences to NCBI SRA and provide the accession number here.

• The authors mentioned about the differential expression of gene 11 upon H.vastatrix infection based on RNA-Seq data but did not provide any supporting data to substantiate the statement. The authors need to give the fold changes and p-value from the RNA-Seq data for all the putative genes identified in this study, and further confirm it by performing qRT-PCR.

• The title is not appropriately reflecting the work done in the manuscript. Most of the structural and functional prediction is the manuscript is done in silico. Thereby, simple mentioning structural and functional analysis will mislead the readers.The author should add in silico to the title to make it clear to the reader.

• Most of the figure legends lack essential information to understand the figure. For example, Figure 1 legend provides the least information. While the title of the figure says about the comparison of the conserved domains, the image is just a snapshot of the NCBI BLAST. And, the red circling does not provide any information about the polymorphism. In Figure 2, the authors mentioned that the protein binding region of gene 5 is boxed in red; instead, they used red underline to mark the region. Same for the gene 11 and green boxes. Authors failed to mention how they come to know that these are the conserved protein binding regions, and based on what? Figure legends should carry enough information about the figure that readers can understand it without digging into the main text.

• Some portions of the material and method section are poorly written. For example, the details about how the phylogenetic tree is generated for figure 4 is not mentioned in the material method section but nicely presented in figure legends. Similarly, authors failed to provide detail information about how the 3D structures of genes 5 and 11 were generated other than merely mentioning the software used for this. Information such as the PDB template used to predict the 3D structure is crucial along with the parameters used for energy minimization. The author should provide enough information about the methods used such that readers can able to replicate the work.

• The authors need to provide synteny maps for the comparison of putative genes with the Canephora and arabica genome.

Minor concerns:

• In Suppl fig. S4, authors need to mark the gene 9-12.

• Inline 270-271, the authors mentioned about two contigs mapped against the transcriptome but only provide images for the contig 9 in suppl fig S4 and wrongly mentioned it fig S5.

• Inline no 190, authors have mentioned that they have tested 21 differential coffee clones using marker CARF005. However, in the FigS2, the gel contains 24 lanes. The authors did not say anything about lane no 22-24 both in figure legends and main text but mentioned it in table 1. The consistency should be maintained. Also, the authors need to label the markers in the gel and specify the size of the PCR band. Same for the gel images in Suppl Fig1b and c.

6. PLOS authors have the option to publish the peer review history of their article (what does this mean?). If published, this will include your full peer review and any attached files.

Reviewer #1: No

Reviewer #2: No

---

## [Author Response · Author response to Decision Letter 0]

3 Apr 2020

Response to reviewers' comments was separately attached.

---

## [Decision Letter · Decision Letter 1]

12 May 2020

PONE-D-19-25123R1

In silico guided structural and functional analysis of genes with potential involvement in resistance to coffee leaf rust: a functional marker based approach

PLOS ONE

Dear Dr. Caixeta,

Thank you for submitting your manuscript to PLOS ONE. After careful consideration, we feel that it has merit but does not fully meet PLOS ONE’s publication criteria as it currently stands. Therefore, we invite you to submit a revised version of the manuscript that addresses the points raised during the review process.

We would appreciate receiving your revised manuscript by Jun 26 2020 11:59PM. To enhance the reproducibility of your results, we recommend that if applicable you deposit your laboratory protocols in protocols.io, where a protocol can be assigned its own identifier (DOI) such that it can be cited independently in the future. For instructions see: http://journals.plos.org/plosone/s/submission-guidelines#loc-laboratory-protocols

We look forward to receiving your revised manuscript.

Kind regards,

Binod Bihari Sahu, Ph.D.

Academic Editor

PLOS ONE

Additional Editor Comments (if provided):

Dear Authors,

Please address the new concerns raised by the reviewers suitably and resubmit.

Thank you

Binod

Reviewers' comments:

Reviewer's Responses to Questions

**Comments to the Author**

1. If the authors have adequately addressed your comments raised in a previous round of review and you feel that this manuscript is now acceptable for publication, you may indicate that here to bypass the “Comments to the Author” section, enter your conflict of interest statement in the “Confidential to Editor” section, and submit your "Accept" recommendation.

Reviewer #1: All comments have been addressed

Reviewer #2: (No Response)

2. Is the manuscript technically sound, and do the data support the conclusions?

Reviewer #1: Yes

Reviewer #2: Partly

3. Has the statistical analysis been performed appropriately and rigorously? 

Reviewer #1: Yes

Reviewer #2: N/A

4. Have the authors made all data underlying the findings in their manuscript fully available?

Reviewer #1: (No Response)

Reviewer #2: Yes

5. Is the manuscript presented in an intelligible fashion and written in standard English?

Reviewer #1: Yes

Reviewer #2: No

6. Review Comments to the Author

Reviewer #1: 1. The authors have addressed the earlier comments. The revised version also includes the supplemental figures.

Reviewer #2: The authors have made an effort to address the questions raised in the first review. However, there are still some grievances that need to be addressed.

Major concerns:

1) The results sections are poorly written, lacking the rationale for the experiment and also the clarity of the experiment. Such as why authors decided to do the gene annotation when they already mentioned genes 5 and 11 are RGAs.

2) And in some cases, there is nothing mentioned about the figure in the result section. For example, nothing is said about figure 3 in the result section.

3) Poor choice of the software for the study. For example, for identifying the conserved domain, the authors used the blastp program instead of the conserved domain database (CDD) of NCBI or similar databases such as SMART that are widely used for this purpose.

4) Poor presentation of the figure. Such as

a. For example, take suppl fig3. It is hard to understand what the authors are trying to share. In figure legend, the authors mentioned the black arrow shows the orientation. In the figure, it is hard to locate the black arrows for gene 11,12 and 10. What is the orange box represents? Again, the poor choice of software/program.

b. In figure 2, the authors failed to mention which one is gene 5 and which one is gene 11.

c. Figure 3: Multiple sequence alignment amino acids were color-coded, which is unnecessary and it makes the underling of amino acid hard to track.

d. The figure quality is not good and pixelated. For example, figure 4 top numbering is hard to follow.

5. Suppl figure4: Authors tried to show the differential expression of genes of contig 9 compared control.

a) The image is of poor quality with hard to read the numbers on top.

b) b. Authors, instead of approximately drawing a rectangle around the gene cluster, should have shown the exact position of the genes.

c) c. Authors also need to provide information about how many transcripts read hit to the genes at different experimental conditions (control vs. different time points) with fold change and p-value in this manuscript instead of directing the reader to their previous publication.

d) Similarly, authors should provide the transcript reads that are matched to the gene 5 of contig three and let readers have that information instead of taking the author's word for granted.

6. The authors hardly cited the figures in their discussion section. Also, the authors failed to discuss what they things could be the reason for gene 11 differentially expressed but not gene 5 when both are RGAs.

Minor concerns:

1. Inline no 113: nanodrop generally provides the quantity and quality of the nucleotides and not the integrity. It needs to be corrected.

2. Authors, in some cases, provided the link to the software they have used in this study and some cases not. Consistency needs to be followed.

3. Inline no 273: authors said CARF005 amplified the region 2065 to 3115 of gene 11, which is of 1050 bps long while the PCR product is around 400 bps. It needs to be corrected.

4. The figure legend of figure 5 is more about how the figure generated rather than what the figure signifies.

7. PLOS authors have the option to publish the peer review history of their article (what does this mean?). If published, this will include your full peer review and any attached files.

Reviewer #1: No

Reviewer #2: No

---

## [Author Response · Author response to Decision Letter 1]

29 May 2020

Response to the comments of the reviewers has been attached along with the other files.

---

## [Decision Letter · Decision Letter 2]

19 Jun 2020

In silico guided structural and functional analysis of genes with potential involvement in resistance to coffee leaf rust: a functional marker based approach

PONE-D-19-25123R2

Dear Dr. Caixeta,

We’re pleased to inform you that your manuscript has been judged scientifically suitable for publication and will be formally accepted for publication once it meets all outstanding technical requirements.

Kind regards,

Binod Bihari Sahu, Ph.D.

Academic Editor

PLOS ONE

Additional Editor Comments (optional):

Reviewers' comments:

Reviewer's Responses to Questions

**Comments to the Author**

1. If the authors have adequately addressed your comments raised in a previous round of review and you feel that this manuscript is now acceptable for publication, you may indicate that here to bypass the “Comments to the Author” section, enter your conflict of interest statement in the “Confidential to Editor” section, and submit your "Accept" recommendation.

Reviewer #1: All comments have been addressed

2. Is the manuscript technically sound, and do the data support the conclusions?

Reviewer #1: Yes

3. Has the statistical analysis been performed appropriately and rigorously? 

Reviewer #1: Yes

4. Have the authors made all data underlying the findings in their manuscript fully available?

Reviewer #1: Yes

5. Is the manuscript presented in an intelligible fashion and written in standard English?

Reviewer #1: Yes

6. Review Comments to the Author

Reviewer #1: (No Response)

7. PLOS authors have the option to publish the peer review history of their article (what does this mean?). If published, this will include your full peer review and any attached files.

Reviewer #1: No

---

## [Editor Report · Acceptance letter]

23 Jun 2020

PONE-D-19-25123R2 

*In silico* guided structural and functional analysis of genes with potential involvement in resistance to coffee leaf rust: a functional marker based approach 

Dear Dr. Caixeta:

I'm pleased to inform you that your manuscript has been deemed suitable for publication in PLOS ONE. Congratulations! Your manuscript is now with our production department. 

Kind regards, 

on behalf of

Dr. Binod Bihari Sahu 

Academic Editor

PLOS ONE